# Demonstrating the Potential of Using Bio-Based Sustainable Polyester Blends for Bone Tissue Engineering Applications

**DOI:** 10.3390/bioengineering9040163

**Published:** 2022-04-06

**Authors:** David H. Ramos-Rodriguez, Samand Pashneh-Tala, Amanpreet Kaur Bains, Robert D. Moorehead, Nikolaos Kassos, Adrian L. Kelly, Thomas E. Paterson, C. Amnael Orozco-Diaz, Andrew A. Gill, Ilida Ortega Asencio

**Affiliations:** 1Mechanisms of Health and Disease, The School of Clinical Dentistry, The University of Sheffield, Sheffield S10 2TA, UK; dhramosrodriguez1@sheffield.ac.uk (D.H.R.-R.); s.pashneh-tala@sheffield.ac.uk (S.P.-T.); amanpreet.bains@medizin.uni-leipzig (A.K.B.); 2Kroto Research Institute, Department of Materials Science and Engineering, The University of Sheffield, Sheffield S3 7HQ, UK; 3The Henry Royce Institute, Department of Materials Science and Engineering, The University of Sheffield, Sir Robert Hadfield Building, Sheffield S1 3JD, UK; r.moorehead@sheffield.ac.uk; 4Polymer IRC, School of Engineering, University of Bradford, Sheffield BD7 1DP, UK; n.kassos@student.bradford.ac.uk (N.K.); a.l.kelly@bradford.ac.uk (A.L.K.); 5Automatic Control and Systems Engineering, University of Sheffield, Sheffield S1 3JD, UK; t.paterson@sheffield.ac.uk; 6Department of Oncology & Metabolism, Medical School, The University of Sheffield, Sheffield S10 2RX, UK; c.orozcodiaz@sheffield.ac.uk; 7Floreon-Transforming Packaging Ltd., Aura Innovation Centre, Bridgehead Business Park, Meadow Rd., Hessle HU13 0GD, UK; andrew.gill@floreon.com

**Keywords:** sustainability, polyester blend, biocompatible, impact strength, bone regeneration

## Abstract

Healthcare applications are known to have a considerable environmental impact and the use of bio-based polymers has emerged as a powerful approach to reduce the carbon footprint in the sector. This research aims to explore the suitability of using a new sustainable polyester blend (Floreon™) as a scaffold directed to aid in musculoskeletal applications. Musculoskeletal problems arise from a wide range of diseases and injuries related to bones and joints. Specifically, bone injuries may result from trauma, cancer, or long-term infections and they are currently considered a major global problem in both developed and developing countries. In this work we have manufactured a series of 3D-printed constructs from a novel biopolymer blend using fused deposition modelling (FDM), and we have modified these materials using a bioceramic (wollastonite, 15% *w*/*w*). We have evaluated their performance in vitro using human dermal fibroblasts and rat mesenchymal stromal cells. The new sustainable blend is biocompatible, showing no differences in cell metabolic activity when compared to PLA controls for periods 1–18 days. Floreon^TM^ blend has proven to be a promising material to be used in bone tissue regeneration as it shows an impact strength in the same range of that shown by native bone (just under 10 kJ/m^2^) and supports an improvement in osteogenic activity when modified with wollastonite.

## 1. Introduction

The use of new and more sustainable polymeric materials is currently of paramount importance and changes in policies favouring the use of bio-based polymers are becoming a reality. Industries and governments are currently driving a change towards sustainability in many fields, and healthcare is no exception. The environmental impact of healthcare-related activities cannot be dismissed as, for example, the NHS in the UK produces more than 400,000 tonnes of waste per year and spends more than £70 m on its disposal [1]. Due to obvious risks of disease transmission and the need for sterility, healthcare waste is routinely incinerated as gold standard practice, however, the reality is that a big percentage of healthcare-related materials could be made of bio-based or sustainable polymers [2]. It is also important to consider that the fabrication of biomaterial devices can also involve sustainable manufacturing, which indeed can contribute to carbon footprint reduction [3]. In this sense, the use of sustainable polymer blends like Floreon™ for the manufacturing of future medical devices is a new approach that we propose to explore in our research. Due to the well-demonstrated superior mechanical properties of the Floreon™ blend, we have focused firstly on exploring the area of bone tissue regeneration.

Bones are the major structural tissue within the human body. Bone acts as support for mechanical loading, provides protection for internal organs, and helps to maintain homeostasis and haematopoiesis [4]. In general, bone tissue possesses a high capacity for self-healing (especially in younger people), and therefore, fractured bones are often able to heal completely without the need for additional intervention. However, large defects are sometimes beyond the regenerative capabilities of bone tissue; these may normally occur as a result of severe trauma or due to the effects of musculoskeletal disease. Over the last few decades, the number of people suffering from musculoskeletal conditions has increased dramatically and bone-related injuries represent a global problem that increasingly affects people’s wellbeing [5,6].

In general, bone fractures that are unable to heal naturally frequently require surgical intervention, as well as the use of bone grafts, implants, and long-term antibiotic treatment [7]. Although autologous bone grafts taken from non-load-bearing sites represent the current gold standard, these are of limited availability and their harvest causes donor site pain and potential donor site infection, among other long-term side effects. Allografts from bone tissue are also available, but they are limited in supply, can carry additional risks of disease transmission (such as hepatitis B and C), and preparation (removal of soft tissues) can compromise their osteogenic and osteoconductive properties [8].

Bone tissue engineering scaffolds offer potential advantages over the drawbacks highlighted above, including availability and endless possibilities of fabrication and design. For bone tissue engineering applications, both the material and design of the scaffold are important to ensure osteointegration, cell proliferation, migration, and differentiation [9]. Material selection in particular requires a deep understanding of bone’s natural composition (a nanocomposite of organic proteins and inorganic materials), mechanical properties, and hierarchical structure [10]. The most common materials used for bone tissue engineering include calcium phosphates in the form of hydroxyapatites, metals such as titanium (and titanium alloys), magnesium, and polyhydroxyalkanoates as well as polylactides [11,12,13,14]. Moreover, scaffold architecture in terms of pore size, volume, and interconnectivity is critical, affecting mechanical performance and the cellular activity within the regenerating bone tissue [15].

Selecting an appropriate manufacturing route for a specific material and design is needed to fabricate bone tissue engineered scaffolds with suitable mechanical and structural properties. Additive manufacturing approaches, including 3D-printing techniques, have become increasingly important for the design of new biomaterial devices. Fused deposition modelling (FDM) is a 3D rapid prototyping technique which uses thermoplastic polymers in the form of filament to deliver multi-layered constructs with intricate structure and tailored shapes. FDM is broadly used in medical applications as well as in dentistry for both the creation of tissue models and the design of custom implants [16]. Although there are a number of well-established polymeric filaments already used in FDM for medical applications, their mechanical properties have yet to be maximised and the development of new biomaterials with improved 3D-printing performance is an emerging area of research.

In this work, we have explored for first time the use of Floreon™ (FLM) for tissue engineering applications, specifically for bone healing. Floreon^TM^ is a biobased and sustainable polymer blend that was recently brought to the market as a new 3D printing filament manufactured using bioresorbable polyesters, including polylactic acid (PLA) and polycaprolactone (PCL); to date, this blend has shown superior mechanical properties and easiness of processing when compared to commercially available counterparts [17]. The inclusion of ceramic materials within polymeric matrices to mimic the composition of natural bone has been shown to improve both mechanical properties and the osteoconductive/osteoinductive nature of bone constructs. In this work we have functionalised the polyester blend via introducing a bioceramic (wollastonite, W). Wollastonite has previously been shown to improve bone regeneration in a comparable way to tricalcium phosphate-based materials via the release of biologically relevant Si ions [18,19,20]. 

To evaluate the potential of Floreon-based bone tissue engineered scaffolds, we have characterized the mechanical properties of 3D-printed Floreon constructs and evaluated their biocompatibility using human dermal fibroblasts and rat mesenchymal stromal cells. The polyester blend has shown to be biocompatible and to have impact strength properties comparable to those presented by native bone. 

The use of sustainable polymers for the fabrication of tissue engineering constructs is a key pathway to achieving healthcare sustainability. The work highlighted in this research is pioneering in the use of Floreon ^TM^ as a sustainable blend for biomaterials design and sets the basis for the development of new and more sustainable 3D-printed constructs for bone regeneration applications, opening the door to the establishment of a more environmentally friendly route to the fabrication of functional scaffolds for bone healing.

## 2. Materials and Methods

### 2.1. Fabrication and Mechanical Characterisation of New Floreon 3D Printing Filament (FLM) and Floreon 3D-Printing Filament Modified with Wollastonite (FLM + W)

The manufacturing process in this work is split into 3 parts. First, the raw materials are compounded into pellets. These pellets are then extruded into a filament suitable for 3D-printing and then this filament is fed into the 3D-printer to manufacture the test specimens used. Mechanical test specimens were produced to determine the elastic modulus, yield strength, and impact strength. These samples were additionally characterised by SEM-EDS to evaluate the surface chemistry and composition as well as DSC to determine the thermal processing characteristics and micro-CT to determine their internal interconnectivity. 

### 2.2. Material Compounding

Three blends were extruded with an APV 19 mm twin screw extruder with L:D ratio of 25:1 with a 1.5 kg/min feeding rate. Floreon™ grade FL785, a proprietary blend of polylactic acid and polycaprolactone at differing molecular weights, was used as the matrix polymer for the study. Wollastonite (Nyglos 8), an acicular grade of wollastonite with an average particle size of 12 µm and an aspect ratio of 19:1 was also provided by Floreon™. All of the materials were dried at 50 °C in a vacuum oven for 24 h prior to processing. 

### 2.3. Material Extrusion

PLA, FLM, and FLM + wollastonite pellets (15% *w*/*w*) were extruded into filaments suitable for 3D-printing by fused deposition modelling (FDM). Filaments were produced using a FilaFab PRO 350 EX extruder (D3D Innovations, UK) fitted with a 2.85 mm nozzle. Extrusion was conducted at 170 °C and 50% drive power. The extruded filament was manually wound onto spools ready for 3D-printing.

### 2.4. 3D Printing of Extruded Material

Mechanical test pieces were 3D-printed by fused deposition modelling (FDM) using an Ultimaker 2+ with nozzle diameter 0.4 mm. The test pieces were designed using CAD software (SolidWorks 2013) and the g-code was generated using Cura (version 3.4.1. USA). The filament was heated to 210 °C for 3D-printing and the build plate was heated to 60 °C. Test pieces were printed with a layer height of 0.1 mm, line width of 0.35 mm, 2 shell layers, 100% infill, and a raster angle of 45°. Build plate adhesion was also specified. Support material was generated, where required, at 10% density. Print speeds were 50, 40, 30, 60, and 40 mm/s for inner walls, outer walls, top/bottom layers, infill and supports, respectively. Following printing, test pieces were finished with 240 grit silicon carbide abrasive paper using a polishing wheel (Buehler Metaserv, Germany). Circular samples, 1 cm in diameter and 0.4 mm thick, for biological testing were also printed following the specifications highlighted above.

### 2.5. Mechanical Testing

A variety of mechanical tests were performed to investigate the performance of 3D-printed Floreon Medical (FLM) with/without the addition of wollastonite (FLM + W). 3D-printed PLA acted as a control material. 

The mechanical test pieces were 3D-printed in three different build orientations to examine the effect this may have on material anisotropy (Figure 1A). The dimensions of the test pieces were described by length (l), width (w), and thickness (t), where l > w > t. Build orientation was described by the parallel relationship between the 3D-printer’s vertical axis (Z) and either of the dimensions l, w, or t. Movement of the printhead in the X and Y axes was considered to be independent and equally free. Therefore, rotation of the builds about the Z axis was considered to have a negligible effect on the material properties of the resulting test pieces. Tensile, compressive, and flexural testing was conducted using a Lloyd Instruments LRX testing machine with Nexygen software (version 4.1). A 2.5 kN load cell was used for all tests. Elastic modulus and yield strength for each material were determined.

Tensile testing was conducted in accordance with the procedure described in BS EN ISO 527-2:2012. The test piece conformed to design Type 1BA (gauge 25 × 5 × 4 mm). Preload was 5N and the test speed was 1 mm/min. The test piece was gripped at each end and loading was applied parallel with its length. Testing proceeded until the stress fell to 50% that of the maximum stress.

Compressive testing was conducted in accordance with the procedure described in BS EN ISO 604-2003. The test piece conformed to design Type A (50 × 10 × 4 mm). The test piece was loaded between two flat plates, both perpendicular to the direction of loading. Preload was 5 N and the test speed was 1 mm/min. Loading was conducted parallel with the length of the test piece. Testing proceeded until the stress fell to 50% that of the maximum stress.

Flexural testing was conducted in accordance with the procedure described in BS EN ISO 178-2010 + A1-2013. The test piece conformed to the preferred specimen type (80 × 10 × 4 mm). Preload was 5 N and the test speed was 2 mm/min. The test piece was suspended across a 64 mm span with its thickness parallel to the direction of loading. Testing proceeded until a deflection of 10 mm was achieved or the load dropped quickly, due to material fracture.

Charpy testing (Tinius Olsen Impact 503) was conducted to determine the impact strength of the 3D-printed materials, in accordance with the procedure described in BS EN ISO 179-1-2010. The test piece conformed to the preferred specimen type (80 × 10 × 4 mm) with a type A milled notch (45°, radius 0.25 mm, depth 2 mm) cut in midway along the length, parallel with the width. The test piece was suspended between a 62 mm span.

### 2.6. Scanning Electron Microscopy (SEM)

3D-printed samples were coated with an 8 nm gold layer using a SC 500A sputter (Emscope). After coating, samples were mounted on aluminium pint stubs (AGG301, Agar scientific) with carbon tabs. Sample imaging was carried out using a HITACHI SEM (FE SEM, JSM-6500F, JEOL and FE/VP SEM, TM3030Plus, HITACHI), spot size was set at 3.5 nm, and voltage was set at 10 kV. ImageJ software v. 1.48 from NIH (National Institutes of Health, USA) [21] was used to analyse the SEM micrographs.

### 2.7. Scanning Electron Microscopy and X-ray Energy Dispersive Spectroscopy (SEM-EDS)

Disk simples of 10 mm in diameter were 3D-printed using one of six materials: PLA, PLA/HA, PLA + W, FLM, FLM/HA, and FLM + W. Each disk was gold-coated in preparation for electron microscopy. SEM images and EDS elemental analysis were performed using a Tescan Vega 3 Scanning Electron Microscope (Tescan Orsay Holding, Czech Republic). Images were taken at 20 kV with 500× magnification.

### 2.8. Differential Scanning Calorimetry (DSC)

Samples of approximately 5 mg were cut from 3D-printed samples of the six materials tested in this study. Differential scanning calorimetry was then performed using a Discovery X3 Differential Scanning Calorimeter (TA Instruments). The test was run on a 10 °C/min ramp from 10 °C to 230 °C. The resulting data was analysed to detect endothermic and exothermic events using Trios V5.0.0.44616 (TA Instruments).

### 2.9. Micro-Computed Tomography (Micro-CT)

3D-printed disc samples were scanned using a Bruker Skyscan 1172 Micro-Computed Tomography. The samples were imaged at 50 kV, 179 µA, a resolution of 11.65 µm, using a 0.5 mm aluminium filter. The raw projections were reconstructed using NRecon (Bruker, Germany).

### 2.10. Biological Testing

#### 2.10.1. Culture of Human Dermal Fibroblasts and Rat Mesenchymal Stromal Cells

Primary human dermal fibroblasts (HDF) were isolated as described by Gosh et al. [22]. The skin was obtained from patients undergoing elective breast reductions and abdominoplasties who gave informed consent for use of their excised skin for research purposes through the Sheffield hospital directorate of Plastic, Reconstructive Hand and Burns surgery research ethics number 15/YH/0177 under the Human Tissue Authority 12179. HDF were cultured in 37 °C incubator at 5% CO_2_ and constant humidity; DMEM AQMedia (Sigma; D0819) was used for cell culture, to which 10% FBS, penicillin-streptomycin, and amphotericin B were also added. The cell medium was changed every 2–3 days until 80% confluence was reached. Circular material discs of PLA, FLM, and FLM + W (15.6 mm in diameter) were placed on 24-well tissue culture plates with 1 mL of DMEM and seeded at a cell density of 10,000 cells/disc. After seeding, samples were incubated at 37 °C, 5% CO_2_, and fed with fresh media intermittently when needed.

Rat mesenchymal stromal cells (RMSC) were extracted from rat bone marrow and stored in liquid nitrogen and DMSO at Passage 0 (P0) until the time of experiment. The cells were thawed and re-suspended in 5 mL of MEM Alpha modification (M4526) (Sigma Aldrich, UK) supplemented with 100 units/mL of penicillin (Sigma Aldrich, UK) and supplemented with penicillin-streptomycin (100 IU/mL–100 µg/mL) (Sigma Aldrich, UK) and centrifuged for 5 min at 1000 RCF and 20 °C. The supernatant was discarded, and pellet was re-suspended in 5 mL of MEM supplemented media. The cell suspension with about 1 million cells was transferred into a T-75 flask (Greiner Bio-One, UK) and incubated at 37 °C and 5% CO_2_ in a humidified atmosphere. The next day, the media in the culture flask was replaced with fresh MEM supplemented media and incubated at 37 °C and 5% CO_2_ in a humidified atmosphere. Cells were passaged when they reached 80% confluency. The cells were seeded on the disc samples placed into a 12-well plated at the cell density of 10,000 cells per sample. The test samples were transferred into a new 12 well plate and fresh supplemented media was added. TCP control was also maintained to assess differences in the cell growth and morphology due to the dye and materials used. After seeding, the cells were incubated at 37 °C, 5% CO_2_, and fed with fresh media every 3 days.

#### 2.10.2. Assessment of Metabolic Activity, Proliferation, and Cell Morphology Using Human Dermal Fibroblasts

##### Metabolic Activity

The metabolic activity of the cells was measured using a resazurin reduction method. The resazurin sodium salt (Sigma; R70117) is a permeable compound that presents a characteristic blue colour before its reduction to resorufin by cell metabolism, thus creating a red colour compound that can be quantified to estimate the number of viable cells. The test was performed at days 1, 3, 5, 10, 15, and 18 for the PLA, FLM, and FLM + W discs seeded with 10,000 cells/mL. A 1 mM resazurin stock solution in PBS was prepared and sterilized using a 0.2 µm membrane syringe filter. The stock solution was used to prepare a 10% (*v*/*v*) working solution using the DMEM culture media. The polymer discs and controls were exposed to 1 mL of the resazurin working solution for 4 h and incubated at 37 °C in a humidified atmosphere with 5% CO_2_. The samples were covered with aluminium foil. After incubation, 150 µL of each sample were loaded into a 96-well plate in triplicate. Absorbance was measured by using a microplate fluorescence reader FLx800 (BIO-TEK instruments) at λex = 540 nm and λem = 630 nm to analyse the change in optical density and thus the amount of resorufin produced.

##### Cell Proliferation

To quantify cell proliferation on the polymer discs, a Invitrogen^TM^ Quant-iT™ PicoGreen™ dsDNA assay kit (P758, Fisher) was performed. Before the assay, a 1× Tris-EDTA (TE) buffer was prepared from the 20 × 200 mM Tris-HCl–20 mM EDTA (pH 7.5) buffer provided by the kit. The 1-fold working PicoGreen^TM^ solution was prepared from the 200-fold stock solution by diluting 100 µL aliquot with 19.9 mL of TE buffer. The Picogreen^TM^ working solution was covered from light using aluminium foil and was kept at 4 °C for no more than 1 month.

Immediately after the cell metabolic activity, samples were washed with PBS, and 500 µL of the TE buffer were added for each well. Next, the plate was covered with parafilm paper on the edges and placed inside a −80 °C freezer for 15 min to induce cell lysis using the freeze–thaw method. Samples were retrieved and defrosted immediately using a water bath at 38 °C. Once completely defrosted, 500 µL of the Picogreen^TM^ working solution were added and incubated for 5 min at RT. After incubation, 200 µL were taken by triplicate per sample and transferred to a 96-well plate. Sample fluorescence was read at λex = 480 nm and λem = 520 nm using a spectrophotometer plate reader (FLx800, Bio-Tek Instruments Inc., USA) and KC4 software (version 3.3, Bio-Tek Inc., USA).

##### Cell Morphology Using Fluorescence and Scanning Electron Microscopy (SEM)

Fibroblast cell morphology on PLA, FLM, and Woll substrates was preliminary studied with fluorescence microscopy. HDF were seeded on PLA, FLM, and WoLL discs placed on 24-well tissue culture plates (CytoOne; CC7672) and then fixed after 5 and 7 days of culturing. Samples were washed with sterile phosphate-buffered saline solution (PBS) and then 1 mL of 3.7% of paraformaldehyde was added and incubated at room temperature for 45 min. Paraformaldehyde was discarded, and samples were washed two times with PBS. Cells were stained with DAPI (4′,6-diamidino-2-phenylindole) and phalloidin TRITC (Tetramethylrhodamine isothiocyanate), analysed under 358 nm and 540 nm, respectively. The staining of the samples was performed by adding 1 mL of 0.1% triton x-100 and incubating at room temperature for 45 min. The triton solution was removed, and the samples were washed three times with PBS. The staining solution was prepared with 10 µL DAPI and 10 µL phalloidin in 10 mL of PBS solution, and 1 mL of the staining solution was added to each sample. The samples were left for 1 h at room temperature covered by aluminium foil. After incubation, samples were washed with PBS and stored at 10 °C. Images were taken using an inverted Olympus fluorescent microscope IX73; samples were imaged under different exposure times of 80–100 ms for DAPI and 850–900 ms for phalloidin TRITC wavelengths. Cell morphology was also analysed using a field emission scanning electronic microscope (FE SEM, JSM-6500F, JEOL). The morphology and topographical cues of the samples were analysed using with the specialized Image J software v.1.48 from NIH (National Institutes of Health, USA). Hexamethyldisilazane (HMDS) was used for sample preparation of polymer discs seeded with human fibroblasts. To preserve cell morphology, formaldehyde 3% in PBS was used for cell fixing. Samples were submerged under 1 mL of 3% formaldehyde for 1 h. After washing the samples with PBS, a dehydration procedure was performed using different solutions of ethanol in distilled water (35%, 60%, 80%, 90%, and 100%), leaving the samples 15 min for each ethanol solution. Samples were treated with a 1:1 ethanol:HMDS solution for 1 h. To complete the process, pure HMDS washes were done to the samples to then allow them to air-dry for 1 h. Samples were then gold coated to create a conductive surface. SEM was performed as previously described for non-seeded samples.

### 2.11. Assessment of Metabolic Activity and Cell Morphology Alkaline Phosphatase Production and Calcium Deposition Using Rat Mesenchymal Stromal Cells

#### 2.11.1. Metabolic Activity

Cell viability of rMSCs on test samples and the controls was assessed using Presto blue^®^ assay according to the manufacturer’s instructions (Invitrogen, Thermofisher Scientific, UK). The assay was performed at Day 1, 3, 7, 14, and 21. 10% presto blue reagent was mixed in fresh supplemented medium. The media from the culture wells containing samples was discarded. 2 mL of 10% presto blue reagent in fresh supplemented media was added to the culture wells. The samples were incubated at 37 °C for 60 min in dark conditions by covering the culture plate with aluminium foil. A media control (MC) consisting of presto blue in fresh supplemented medium without cells was also maintained. After incubation, the 200 μL (×2) of presto blue media from each well was removed and transferred into a 96 well plate (Greiner 96 flat transparent) (Sigma Aldrich, UK) and stored in dark conditions until the absorbance was read. Cell viability was assessed based on the fluorescence intensity. Fluorescence intensity was measured using Tecan infinite M200 plate reader and Magellan data analysis software (Tecan 2013) set up with the presto blue assay parameters in accordance with the manufacturer’s instructions. After completing the assay, cells were fixed by removing the remaining presto blue solution and incubating the cells with 3.7% paraformaldehyde for 1 h at RT.

#### 2.11.2. Cell Morphology

Fixed cells were stained with phalloidin and DAPI (as described above) to study cell morphology via F-actin and cell nuclei observations, respectively. Stained cells were observed under an inverted Olympus fluorescent microscope IX73.

#### 2.11.3. Assessment of Osteogenic Potential Using Alkaline Phosphatase and Calcium Deposition Assays Using Alizarin Red Staining

The osteogenic potential of the rMSCs on different samples was estimated using alkaline phosphatase assay. ALP assay was performed on the freeze thawed samples. Phosphatase substrate (Sigma Aldrich, UK), Alkaline phosphatase buffer (1.0 M Diethanolamine and 0.50 mM Magnesium chloride) (Sigma Aldrich, UK), and 10 mM p-Nitro phenyl Phosphatase (pNPP) (Sigma Aldrich, UK) were used to determine the osteogenic potential. The phosphatase substrate was prepared by dissolving 6.6 mg of substrate in 1 mL of filtered, distilled water. A standard curve was obtained by serially diluting pNPP (10 nmole/mL) solution. 50 μL of sample solution (×3) (samples and the serially diluted pNPP solution) was transferred to a 96 well plate. 190 μLof assay buffer and 10 μL of substrate was added to each well. The samples were incubated at room temperature for 45 min (time can be varied depending on the type of sample) in dark conditions by covering the 96 well plate with an aluminium foil. The absorbance was measured using Tecan infinite M200 plate reader and Magellan data analysis software (Tecan 2013) set up with the alkaline phosphatase assay parameters in accordance with the manufacturer’s instructions. The scaffolds and TCP controls were washed with PBS twice. 

The ALP activity data was normalized using picogreen data to obtain ALP activity (nmole·min^−1^) per ng/mL of DNA. After culturing the rMSCs on the test samples and TCP, the samples were analysed for calcium deposition using Alizarin Red staining. For this, 40 mM Alizarin red satin was prepared by dissolving 0.136 gms of Alizarin red stain (Sigma Aldrich UK) in distilled water. The cell culture medium was removed, and the samples were washed twice with 2 mL of PBS for 2 min. After washing, the cells were treated with 4% paraformaldehyde in PBS (Alfa Aesar, Thermofisher Scientific, UK) for 1 h. After treatment, the samples were washed three times with PBS followed by a rinse in distilled water for 5 min. 0.5 mL of alizarin stain solution was added to the culture wells and incubated for 15 min at room temperature. After incubation, the excess stain was removed by washing the samples several times with distilled water. After the excess stain was removed, the samples were imaged using a digital camera.

### 2.12. Statistical Analysis

GraphPad Prism software (version 9.1, USA) was used to perform statistical analyses using one-way or two-way analysis of variance (ANOVA) as applicable, followed by Tukey’s multiple comparisons tests. In all cases, *p* values < 0.05 were considered statistically significant.

## 3. Results

Comparing the different materials under tensile loading, the data show that PLA was significantly stiffer (based on E modulus) than FLM or FLM + W in the l-Z and w-Z orientations (Figure 1B). There was no significant difference between PLA, FLM, and FLM + W when printed in the t-Z orientation. Comparing print orientations, PLA was significantly less stiff when printed in t-Z, with no difference observed between the l-Z and w-Z orientations. Print orientation had no significant effect on FLM. FLM + W printed in the t-Z orientation was stiffer than in l-Z. These results are partially mirrored in the yield strength data (Figure 1C), with PLA showing a significantly higher yield strength than FLM and FLM + W when printed in l-Z or w-Z, but not t-Z. However, when comparing print orientation, PLA printed in w-Z appeared stronger compared to l-Z. In both FLM and FLM + W, printing in l-Z resulted in significantly lower yield strengths compared to w-Z and t-Z, which were not significantly different from each other. Loading in compression showed that PLA produced significantly stiffer structures than FLM or FLM + W when printed in l-Z and t-Z orientations (Figure 1B). There was no significant difference in E modulus in compression between the materials when printed in the w-Z orientation. Print orientation appeared to have little effect when considering the materials individually, with only printing of FLM in l-Z and w-Z producing E modulus values in compression that were significantly different. 

Again, these results are partially mirrored in the data for yield strength in compression, with PLA generating significantly higher values than FLM or FLM + W when printed in the l-Z and t-Z orientations, but not w-Z (Figure 1C). There was no effect of print orientation on the compressive yield strength of PLA, however, printing in the w-Z orientation produced significantly greater compressive yield strength values for FLM and FLM + W materials. 

When loaded in bending, PLA was significantly stiffer than FLM in l-Z and t-Z print orientations and FLM + W was significantly stiffer than FLM in the l-Z print orientation. There was no difference between the stiffness of the materials in bending when printed in the w-Z orientation. Comparing the materials in different print orientations, significant differences in stiffness in bending were only observed in FLM when comparing l-Z and w-Z orientations.

Considering yield strength in bending, PLA produced significantly higher values than FLM and FLM + W in all print orientations. Additionally, printing in w-Z and t-Z orientations produced significantly higher values than in l-Z in all materials.

Impact testing revealed that the impact strength of PLA was significantly lower than FLM or FLM + W in all print orientations (Figure 1D). FLM and FLM + W were not significantly different at equivalent print orientations. Print orientation did not significantly affect the impact strength of PLA. Print orientation t-Z produced the greatest impact strength values in FLM and FLM + W. 

Chemical analysis demonstrated that all samples showed peaks corresponding to C, O, and Au, which were expected from the polymer matrix and the coating for microscopy. Unexpectedly, all samples also showed detectable levels of Ca and Si, including those printed on pure PLA and pure Floreon (Figure 2); as explained in the discussion, we believe this was due to contamination.

All materials displayed three distinct events during DSC characterisation: an early endothermic peak (glass transition), an exothermic peak (crystallisation), and a late endothermic peak (melting) (Figure 3A). For the PLA materials, the first endothermic peak corresponds to its glass transition temperature, for which the onset point could not be detected. The exothermic peak corresponds to its crystallisation temperature, and the second endothermic peak corresponds to its melting temperature. The Floreon materials displayed a similar behaviour, but with a better defined first endothermic peak, for which an onset could be clearly detected.

Micro-computed tomography images (Figure 3C) show radio-opaque particles within the volume of the FLM-W samples, but not on the PLA and FLM ones. These would correspond to calcium-containing micro-scale ceramic agglomerations: the absence in the pure materials suggests there are only trace levels of ceramics in these, probably from contamination of the processing machinery. 

HDF cell viability across all tested materials was significantly lower only for PLA discs at days 3 and 15 compared to the TCP (tissue culture plate control group). FLM and FLM + W discs showed similar or higher absorbance values compared to the TCP group (Figure 4A). Moreover, there was no significant difference in cell viability between the FLM and FLM + W groups. The Picrogreen™ assay (Figure 4B) showed that there was no difference in cell proliferation among groups either at days 1 or 18. This indicates that the substrate (PLA, FLM, or FLM + W) only influenced cell metabolic activity for HDF. This behaviour was also observed under the microscope, as the three polymer surfaces showed HDF confluency. The fabrication method used to print the discs created microtopographical cues observed as defined squares due to the deposition of ECM. Topographical cues are relevant on fibroblast matrix deposition and migration [23,24], but also have proved to be critical on osteoblast differentiation. Therefore, more studies are needed to understand their effects [25].

## 4. Discussion

Our research has focused on testing the viability of using a new sustainable polyester blend for the development of 3D-printed constructs for bone tissue engineering applications. The use of polyester-based materials and FDM approaches has proven to be successful for the development of biomaterials with bone regenerative capabilities and many FDM/polyester-based promising constructs can be found in the literature, generally showing superior regeneration capabilities when compared to autografts (PCL-BMP-doped scaffolds [26] as well as PCL/tricalcium phosphate scaffolds produced by FDM [27] are some examples). 

One of the aims of this project was to prove the suitability of Floreon™ polyester blend for its use in tissue engineering applications, particularly for bone regeneration. For this, our first approach was to produce a comprehensive study of the mechanical properties of 3D-printed Floreon™ samples, comparing them to PLA controls as well as to 3D-printed Floreon modified with a bioceramic (Wollastonite). In essence, our mechanical testing results showed a significantly higher degree of stiffness for the PLA samples across tension, compression, and bending when compared to FLM and FLM + W samples. This was expected, as the inclusion of polycaprolactone within the Floreon blend makes this material softer than regular PLA. However, when analysing the data for impact strength, Floreon™ showed distinctly superior properties compared to PLA. This is a key finding, as “impact strength” is regarded as one of the most relevant fracture properties to consider when developing bone substitutes [28]. 

Typical impact strength values for fresh bone have been shown to have average values just below the 20 kJ/m^2^, as reported by W.C. Hayes and co-workers [29,30]. In this research we have demonstrated that 3D-printed PLA constructs show themselves a considerable impact strength with average values just under 3 kJ/m^2^ (see Figure 1), however, Floreon samples are able to more than triple these (getting close to 10 kJ/m^2^), approaching that of human compact bone. This finding demonstrates the potential of our sustainable 3D-printed material to be a very good candidate for the development of scaffolds for bone regeneration. Another aspect that we have explored in this work is the fact that printing orientation can have a significant impact on the mechanical properties of the manufactured 3D-printed constructs. This has been widely reported and our data confirms previously published trends [31,32]. Specifically, we have observed that bonding within layers seems to be weakened by the presence of the ceramic component for some of the orientations and types of testing (see Figure 1). Issues with processing ceramic-modified FDM filaments have been widely reported, both in terms of surface roughness control as well as printed orientation, which is normally predetermined by the size of the extruded filament and results in limited control in the z direction [33]. Moreover, as a general trend, our results show that Floreon™ materials are less stiff when printed in the l-Z direction.

In terms of composition, the ceramic biofunctionalised materials showed clear presence of Si, as expected from the ceramic additive. Pure-PLA and pure-Floreon samples also showed traces of Si and this was likely due to contamination from the filament extruder and the 3D-printer, despite both being purged before use. Pure polymer samples also exhibited strong signals for Na and Cl. This may be from salts dissolved in water used during manufacturing of the raw PLA, or residues of desiccants used for storage. This contamination is not present on the samples with wollastonite. Pure PLA also showed signals for Cd and K and these could be residues for catalysts commonly used during lactide polymerisation. From the micro-CT data, it would appear that most of the wollastonite is aggregated in micron-size particles, which would not be clearly visualized at this resolution.

The use of human skin types to assess preliminary in vitro cell viability and proliferation for bone tissue engineering applications has been proposed before as an effective platform [34] due to their secretory role on bone regeneration by fibroblasts growth factors [35]. However, most of the work with dermal fibroblasts and bone tissue engineering studied the effect of the polymer substrate on cell matrix formation, especially in presence of a mineral substrate [36,37,38]. Claeys et al. underlined the efficiency of dermal fibroblasts as a model to study bone disorders, as well as their potential for bone regeneration studies by inducing osteogenic differentiation [39]. The results of our work showed that in vitro fibroblast viability and proliferation is not hindered when cultured on FLM or FLM-W and that FLM results are comparable to TCP and PLA controls for periods up to 18 days (Figure 5). 

Mesenchymal stromal cells were also seeded on the Floreon™ scaffolds and they were found to attach to our membranes and proliferate as expected (see Figure 5A). When measuring ALP production, the polyester blend modified with wollastonite did support higher osteogenic differentiation. Previous studies have also employed wollastonite as an active agent in polymer scaffolds for bone tissue regeneration. The addition of W has been associated with enhanced bone formation, producing equal or superior results to tricalcium phosphate [18,20,40,41]. This is believed to be a result of the action of the Si ions released from the bioceramic, which is known to stimulate osteoblast proliferation and differentiation and to enhance angiogenesis [42,43,44,45]. Additionally, when used in conjunction with degradable polymer scaffolds, wollastonite may also provide a buffering effect which positively impacts tissue regeneration. The degradation products of many polymer scaffolds can cause the acidification of the local environment, leading to inflammation and fibrosis; W may act against this, maintaining pH and thus improving tissue growth [20]. The combination of wollastonite and polymer may also help to prolong the positive action of the ceramic material. Wollastonite alone dissolves quickly in vivo, resulting in its local bioactivity being short-lived. The slower degradation of polymer scaffolds compared to W alone enables the mineral to persist within the tissue, increasing the duration of its action [46]. Alizarin red staining was also used alongside ALP studies and our results confirmed that an early bone-like matrix was formed after 28 days in culture, indicating the presence of deposits of calcium (typically found in differentiated osteoblasts) [47]. Control groups did not present deposition of calcium.

In essence, mechanical characterisation as well as tissue culture studies have demonstrated that Floreon™ scaffolds have the potential to be used as bone regeneration constructs, both mimicking the mechanical strength of human bone and also providing a biologically relevant environment for MSC’s differentiation.

## 5. Conclusions

In this research we report, for the first time, the use of Floreon™ polyester blends for tissue engineering applications, specifically for bone regeneration. An exhaustive study of the mechanical properties of Floreon 3D-printed constructs (comparing them to counterpart PLA) has demonstrated their suitability to be used as bone regeneration scaffolds as they present higher impact strength than PLA, showing values in the order of those presented by natural bone. This study is pioneering in showing that 3D-Printed Floreon™ material is biocompatible, and this has been studied using dermal fibroblasts and mesenchymal stromal cells. Moreover, modification of the material with a bioceramic (wollastonite) has indicated that Floreon–wollastonite blends are promising osteoblastic differentiation biomaterials. In essence, we have demonstrated that the new sustainable Floreon^TM^ blend is biocompatible and has the potential to act as a bone regeneration platform.

## Figures and Tables

**Figure 1 bioengineering-09-00163-f001:**
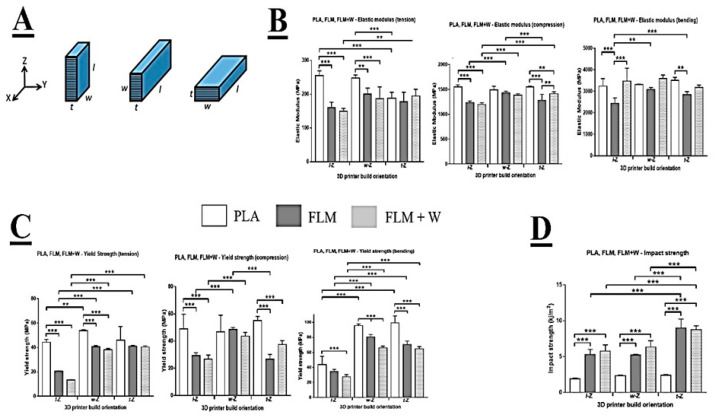
Mechanical analysis of 3D-printed samples. Diagram of printing orientation (**A**). Comparison between elastic modulus (tension, compression, and bending) across different printing orientations (**B**). Comparison between yield strength (tension, compression, and bending) across different printing orientations (**C**). Changes in impact strength for different printing orientations (**D**). All mechanical tests were performed for PLA, FLM, and FLM + W samples. Results are shown as mean ± SD *** *p* < 0.001, ** *p* < 0.01. N = 2, n = 3.

**Figure 2 bioengineering-09-00163-f002:**
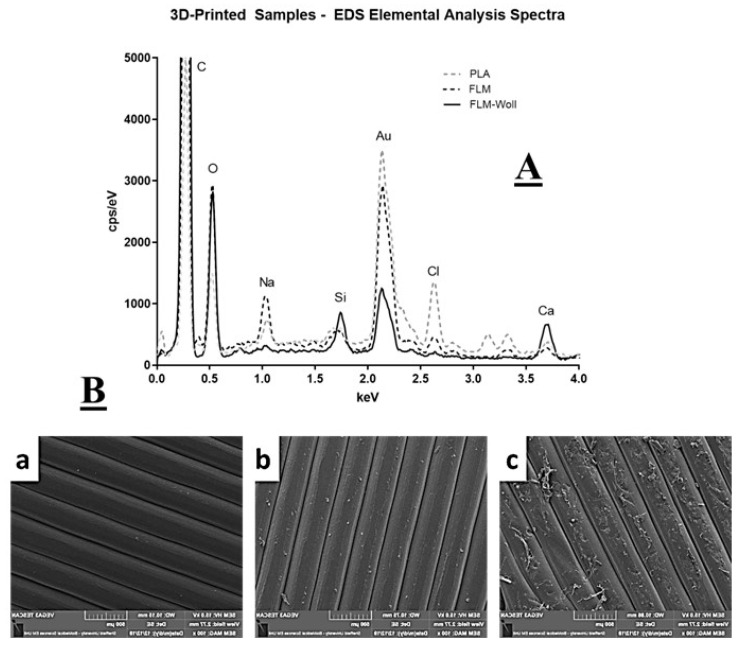
(**A**) Elemental analysis results for PLA, FLM, and FLM-W. A large peak was detected (and cropped) for carbon in all samples, which was expected as the main polymer matrix of the samples. Similarly, a peak for oxygen was also detected, as well as a prominent peak for gold from the sample preparation for electron microscopy. Expected peaks for calcium were also detected in wollastonite samples. Some Ca peaks were also detected on the PLA and FLM samples; this appears to have been only surface contamination. Other types of contamination were also detected in all samples, including silicon, chlorine, and sodium, which could be attributed to material processing contamination. (**B**) Electron microscopy images on the surface of 3D-printed samples of PLA (**a**), FLM (**b**), and FLM-W (**c**). Imperfections detected on the surface of FLM-W are likely to be caused by the presence of ceramic particles within the polymer matrix.

**Figure 3 bioengineering-09-00163-f003:**
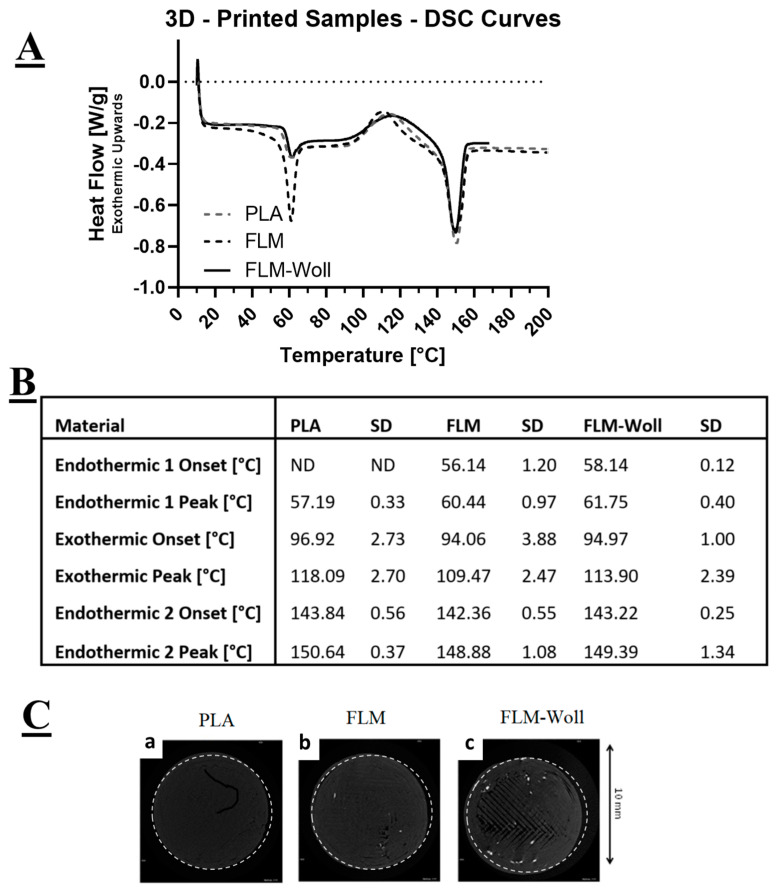
(**A**) DSC curves for PLA, FLM, and FLM-W. The addition of wollastonite was not found to affect the overall thermal behaviour of the material, or (**B**) its critical temperatures. (**C**) Micro-computed tomography images for PLA, FLM, and FLM-W showing ceramic distribution within the 3D-printed constructs.

**Figure 4 bioengineering-09-00163-f004:**
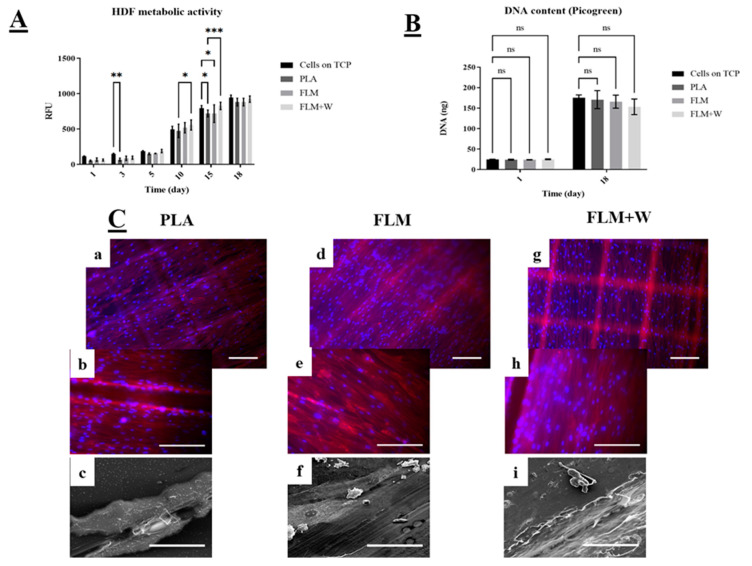
Study of HDF in vitro cell viability and proliferation. Resazurin assay (**A**) showed FML and FML + W substrates induced higher values of resazurin reduction compared to the TCP controls. PicoGreen™ DNA quantification assay (**B**) also indicates that the substrates did not have a negative effect on cell proliferation. Results are shown as mean ± SD *** *p* < 0.001, ** *p* < 0.01, * *p* < 0.05, ns *p* ≥ 0.05. N = 3, n = 3. (**C**) Micrographs taken on day 18 using fluorescence microscopy with DAPI (blue) and phalloidin-TRITC (red) to stain the nucleus and cytoskeleton, respectively. Images (**a**,**b**) show HDF growing on PLA controls, images (**d**,**e**) show HDF growing on FLM surfaces and images (**g**,**h**) show HDF growing on FLM+W samples (Scale bar = 200 µm); SEM images (**c**,**f**,**i**) show matrix deposition on the surface of the polymer discs.

**Figure 5 bioengineering-09-00163-f005:**
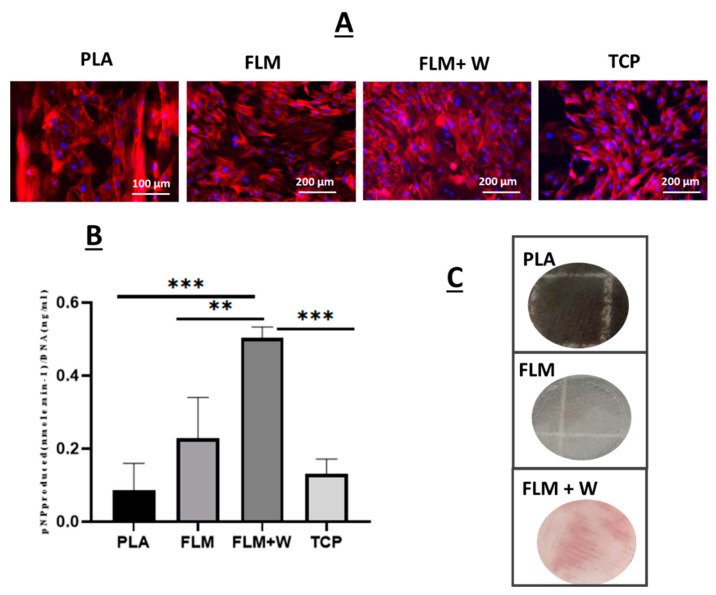
Panel (**A**) shows fluorescence images of rat mesenchymal stromal cells (MSCs) seeded on tissue culture plastic controls (TCP), PLA, and FLM/FLM + W samples for 14 days. Actin filaments are shown in red (Phalloidin TRIT-C) and nuclei in blue (DAPI). Panel (**B**) shows ALP data measured at 14 days of culture for TCP and PLA controls, as well as for FLM and FLM + W samples. Samples with wollastonite presented significant differences for ALP expression with respect to all the other groups (*** *p* < 0.001, ** *p* < 0.01); (**C**) Alizarin red optical images show calcium deposition for the MSC cells seeded on FLM + Woll samples.

## Data Availability

Not applicable.

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
