# Peer review of "Demonstrating the Potential of Using Bio-Based Sustainable Polyester Blends for Bone Tissue Engineering Applications"

_bioengineering, 2022, doi:10.3390/bioengineering9040163_

Round 1
Reviewer 1 Report
- Quantitative data is missing in the abstract.
- Discuss the individual reference in the introduction section.
- Highlight the novelty of research work and research gaps of literature review.
- Highlight the major contribution in the conclusion section.
The manuscript is well written and prested. I recommend for consideration after information of minor comments.
Author Response
1. Quantitative data is missing in the abstract.
The authors would like to thank the reviewer for spotting this issue. We have now added quantitative data in the abstract (lines 29-35).
2. Discuss the individual reference in the introduction section.
Thank you for highlighting this as it was a mistake. References 1, 2, 3 were discussed separately in the intro section but then they were grouped as (1-3) for another claim. The grouped references were incorrect, an error when formatting. The (1-3) reference group in the Introduction has now been deleted (line 53).
3. Highlight the novelty of research work and research gaps of literature review.
We agree this was not clearly highlighted. The last paragraph for the Intro has now been modified to further highlight the gap in knowledge and to explain why our study is pioneering (lines 111-123).
4. Highlight the major contribution in the conclusion section.
We agree, major contributions were not clear enough in the conclusion; these have been clarified now in the revised version (lines 590-602).
The manuscript is well written and prested. I recommend for consideration after information of minor comments.
Reviewer 2 Report
The authors present here a manuscript discussing the use of bio-based sustainable polyester blends for bone tissue engineering applications. The manuscript is generally well written. The results are consistent and presented in a logical progression.
However, several aspects of the manuscript could be improved:
- microCT data regarding interconnectivity are not exposed
- stability, toxicity and resorbability data are not investigated and should be reported here
- in vivo integration of all sample compositions should be investigated in a bone defect model (bearing or non-bearing bone)
Minor:
- Figure 1A is not cited in the text of the manuscript
Author Response
The authors present here a manuscript discussing the use of bio-based sustainable polyester blends for bone tissue engineering applications. The manuscript is generally well written. The results are consistent and presented in a logical progression.
However, several aspects of the manuscript could be improved:
- microCT data regarding interconnectivity are not exposed: The micro-CT study was performed to gather data related to the ceramic material location within the polymeric matrix. Our samples were not manufactured to be deliberately porous (as the material is molten during 3D-printing manufacturing and any trapped bubbles are not intentional). At the scanned resolution (11.65 um), any porosity would need to be at least around 50 microns to be properly analysed using our micro-CT set up (this includes pore and element interconnectivity). In essence, any irregularities on density would be macroscopic and not related to material microstructure. For such an analysis on our material, a nano-CT level resolution would be required. However, we do acknowledge that micro-CT data was not clearly cited within the text body for the results section and we have now modified this to make it clearer (lines 453-468).
- stability, toxicity and resorbability data are not investigated and should be reported here: The biodegradability of the material is a very important point indeed and we are currently studying this in our laboratory both for 3D-printed constructs made of Floreon as well as for fibrous scaffolds. This manuscript reports the use of Floreon for first time in tissue engineering and focuses on its suitability in bone due to its mechanical properties; although we agree these points are key when working towards the use of the material for a future clinical application, these experiments would be outside the scope of the presented manuscript.
- in vivo integration of all sample compositions should be investigated in a bone defect model (bearing or non-bearing bone): Thank you as well for highlighting this point; our next steps will include planning for the use of in vivo models and for attracting funding to ensure their feasibility; however, these are out of the scope of this particular paper which aimed to explore and demonstrate the feasibility of using Floreon sustainable blends for bone tissue engineering applications.
Minor:
- Figure 1A is not cited in the text of the manuscript
Thank you very much for highlighting this point; figure 1A is now mentioned in line 167 and a further mention for figure 1D has been included in line 413 too (to add clarify to the last part of the relevant results paragraph).
Reviewer 3 Report
In this study, the authors have investigated the suitability of using a new sustainable polyester blend (Floreon™) as a scaffold for application in bone tissue engineering. They developed a series of 3D-printed constructs from a novel biopolymer blend using fused deposition modelling (FDM), and modified these materials using wollastonite. Finally, they have evaluated the in vitro performance of the designed materials using human dermal fibroblasts and rat mesenchymal stromal cells. The study is interesting and reports interesting results; however, there are minor issues need to be addressed before the manuscript can be reconsidered for publication.
1-Introduction: more background on biocompatibility and biomineralization are necessary.
2- It has been reported in literature that wollastonite has a toxic effect on red blood cells. So, the reviewer recommends performing in vitro blood compatibility analysis (hemolysis assay; if applicable).
3-The best cells for the in vitro biocompatibility evaluation are those cells that will potentially come in contact with the materials. Since, your material is designed for bone engineering. Therefore, osteoblast and endothelial cells are the best choices. Why did not the authors use osteoblast cells or another kind of bone cells for cytotoxicity evaluation instead of dermal fibroblast?
4-In lines 229 and 231, correct 37â—¦C.
Author Response
In this study, the authors have investigated the suitability of using a new sustainable polyester blend (Floreon™) as a scaffold for application in bone tissue engineering. They developed a series of 3D-printed constructs from a novel biopolymer blend using fused deposition modelling (FDM) and modified these materials using wollastonite. Finally, they have evaluated the in vitro performance of the designed materials using human dermal fibroblasts and rat mesenchymal stromal cells. The study is interesting and reports interesting results; however, there are minor issues need to be addressed before the manuscript can be reconsidered for publication.
1.Introduction: more background on biocompatibility and biomineralization are necessary.
Thank you for highlighting this issue; more background related to the use of bioceramics has been now added as a new paragraph in the introduction (lines 102-109).
2- It has been reported in literature that wollastonite has a toxic effect on red blood cells. So, the reviewer recommends performing in vitro blood compatibility analysis (hemolysis assay; if applicable).
We really valuate this insight from the reviewer; although performing blood compatibility analysis is out of the scope of this particular paper, we will take the advice into account for our future work and we will explore the feasibility of performing blood compatibility tests to ensure the safety of the proposed constructs.
3-The best cells for the in vitro biocompatibility evaluation are those cells that will potentially come in contact with the materials. Since, your material is designed for bone engineering. Therefore, osteoblast and endothelial cells are the best choices. Why did not the authors use osteoblast cells or another kind of bone cells for cytotoxicity evaluation instead of dermal fibroblast?
Thank you for highlighting this issue; as pointed in the submitted paper (lines 553-562) dermal fibroblasts have been widely used for bone regeneration studies. In our particular case, we wanted to perform an evaluation for FloreonTM to be used in tissue regeneration (in general), that is why we started our research approach using dermal fibroblasts; moreover, we started our work with this particular cells because they are primary and they have been proven to offer very valuable/reliable information when used in our laboratory testing other materials (rather than using a commercial immortalised cell line). When we demonstrated that the mechanical properties of 3D-printed Floreon were suitable for it to be explored for bone healing, we then used mesenchymal stromal cells to specifically explore osteogenic potential.
4-In lines 229 and 231, correct 37â—¦C.
Thank you very much for spotting this formatting error that has now been resolved (please see revised manuscript lines 245&247).
Reviewer 4 Report
The article titled Demonstrating the potential of using bio-based sustainable polyester blends for bone tissue engineering applications aimed at showing the potential of polyester supplemented with wollastonite for bone engineering application. The article is well written and the author put effort into making it clear. There are some comments regarding:
- the materials and methods where some typos are listed below, but this is not a complete list so the manuscript should be double-checked.
- It is clear that the authors know that the phalloidin is an actin cytoskeleton marker, but it would be interesting to discuss the differences of the shape of the cytoskeleton between the different conditions and its orientation compared to the orientation of the fibres of the scaffold. Indeed it is well known that the physical alterations of the scaffold are changing cells' behaviour.
- For clarity, it would be easier to have less description of the non significant results. It is difficult to understand, from the point of view of the author, which parameter is the most important here when designing scaffolds for bone engineering.
- In addition, were mechanical tests performed post cells seeding and at the end of the culture? It would have being interesting to see if the deposit of calcium are affecting already the mechanical properties.
- Discussion: it is not clear what is the advantage of using the wallonite compared to other components already used in bone engineering, and in terms of osteogenesis. Is this component inducing a faster differentiation compared to another biomineral?
Minor comments:
- Some typo between lines 215 and 221: extra full-stop and bracket, the 2 of CO2 is not in lower case
- Based on which parameter was the medium changed for the HDF? It is not clear what means "intermittently when needed".
- The cells were not deforested, but thawed
- Please check the concentration for the streptomycin is the right one as it is unusual
- Figure 4: the phalloidin doesn't label the body of cells, it does stain the cytoskeleton.
- Figure5: it is a deposition of calcium not a formation, and it would be more practical for the reader that the organisation of the chart B and the images A are matching orders.
Author Response
The article titled Demonstrating the potential of using bio-based sustainable polyester blends for bone tissue engineering applications aimed at showing the potential of polyester supplemented with wollastonite for bone engineering application. The article is well written and the author put effort into making it clear. There are some comments regarding:
The materials and methods where some typos are listed below, but this is not a complete list so the manuscript should be double-checked.
Thank you for pointing this out; we have gone through the section and corrected the typos the reviewer has highlighted and any other mistake that we were able to spot. We have proof-read not only Materials and Methods but the entire manuscript (see corrections throughout the revised manuscript).
It is clear that the authors know that the phalloidin is an actin cytoskeleton marker, but it would be interesting to discuss the differences of the shape of the cytoskeleton between the different conditions and its orientation compared to the orientation of the fibres of the scaffold. Indeed it is well known that the physical alterations of the scaffold are changing cells' behaviour.
Thank you for the comments; we believe it might have been a misunderstanding from the reviewer for this particular point as the scaffolds we present here are not fibrous. The scaffolds are 3D-printed constructs, maybe the author refers to the layer-by-layer effect when manufacturing the samples with fused deposition modelling as these layers can be identified in some of the presented images. If that is the case, yes, we agree that a future project could focus on studying cell behaviour at the interface of those layers because as the reviewer has highlighted the physical alterations could have an effect on cell behaviour.
For clarity, it would be easier to have less description of the non-significant results. It is difficult to understand, from the point of view of the author, which parameter is the most important here when designing scaffolds for bone engineering.
Thank you for the advice, we agree the most important highlights for the paper were not clearly explained (this was pointed by reviewer 1, see above) and, accordingly, changes in the introduction and conclusion have been incorporated in the revised version for these sections.
In addition, were mechanical tests performed post cells seeding and at the end of the culture? It would have being interesting to see if the deposit of calcium are affecting already the mechanical properties.
This is a very interesting point and we are grateful the reviewer has highlighted it. No, we did not perform mechanical testing after cell culture but we agree it would be a very interesting thing to study and we will incorporate this milestone in our current project which does involve the use of fibrous membranes, Floreon and Wollastonite.
Discussion: it is not clear what is the advantage of using the wallonite compared to other components already used in bone engineering, and in terms of osteogenesis. Is this component inducing a faster differentiation compared to another biomineral?
In the discussion it is mentioned that Wollastonite can perform with an equal and/or higher performance than commonly used osteogenic materials such as tricalcium phosphate (567-580); a new short paragraph clarifying the use of wollastonite in our material has been added in the intro section (as this issue was also pointed by reviewer 3 (lines 102-109).
Minor comments:
Some typo between lines 215 and 221: extra full-stop and bracket, the 2 of CO2 is not in lower case
Thank you very much, this has now been sorted.
Based on which parameter was the medium changed for the HDF? It is not clear what means "intermittently when needed".
Media was changed every 3 days; this has now been corrected (line 253).
The cells were not deforested, but thawed
Thank you very much, this has been corrected now (line 239).
Please check the concentration for the streptomycin is the right one as it is unusual
Thank you very much for pointing this out as the concentration was wrong and it has now been corrected, please see revised manuscript (line 241).
Figure 4: the phalloidin doesn't label the body of cells, it does stain the cytoskeleton.
Thanks, this has been corrected now (see figure 4 caption in revised version, line 489).
Figure5: it is a deposition of calcium not a formation, and it would be more practical for the reader that the organisation of the chart B and the images A are matching orders.
Thank you very much for raising these important points. “Formation” has now been changed by the correct word “Deposition” in the figure caption (see figure 5 caption in revised manuscript, line 502). Moreover, the order of the images in panel “A” has been changed to match panel “B” as the reviewer suggested (please see new figure in the revised version of the manuscript, line 495).